# Fast-Dissolving Nifedipine and Atorvastatin Calcium Electrospun Nanofibers as a Potential Buccal Delivery System

**DOI:** 10.3390/pharmaceutics14020358

**Published:** 2022-02-04

**Authors:** Hassa A. Alshaya, Ahmed J. Alfahad, Fatemah M. Alsulaihem, Alhassan H. Aodah, Abdullah A. Alshehri, Fahad A. Almughem, Haya A. Alfassam, Ahmad M. Aldossary, Abdulrahman A. Halwani, Haitham A. Bukhary, Moutaz Y. Badr, Salam Massadeh, Manal Alaamery, Essam A. Tawfik

**Affiliations:** 1National Center of Biotechnology, Life Science and Environment Research Institute, King Abdulaziz City for Science and Technology (KACST), Riyadh 11442, Saudi Arabia; halshaya@kacst.edu.sa (H.A.A.); ajlfahad@kacst.edu.sa (A.J.A.); falsulaihem@kacst.edu.sa (F.M.A.); aaodah@kacst.edu.sa (A.H.A.); abdualshehri@kacst.edu.sa (A.A.A.); falmughem@kacst.edu.sa (F.A.A.); aaldossary@kacst.edu.sa (A.M.A.); 2KACST-BWH Centre of Excellence for Biomedicine, Joint Centers of Excellence Program, King Abdulaziz City for Science and Technology (KACST), Riyadh 11442, Saudi Arabia; halfassam@kacst.edu.sa (H.A.A.); massadehsa@ngha.med.sa (S.M.); alaameryma@ngha.med.sa (M.A.); 3Pharmaceutics Department, Faculty of Pharmacy, King Abdulaziz University, Jeddah 22254, Saudi Arabia; aahalwani@kau.edu.sa; 4Regenerative Medicine Unit, King Fahd Medical Research Center, King Abdulaziz University, Jeddah 22254, Saudi Arabia; 5Department of Pharmaceutics, College of Pharmacy, Umm Al-Qura University, Makkah 24382, Saudi Arabia; habukhary@uqu.edu.sa (H.A.B.); mybadr@uqu.edu.sa (M.Y.B.); 6Developmental Medicine Department, King Abdullah International Medical Research Center, King Saud Bin Abdulaziz University for Health Sciences, King Abdulaziz Medical City, Ministry of National Guard Health Affairs (MNGHA), Riyadh 11481, Saudi Arabia; 7Saudi Human Genome Satellite Laboratory at King Abdulaziz Medical City, King Abdulaziz City for Science and Technology (KACST), Ministry of National Guard Health Affairs (MNGHA), Riyadh 11481, Saudi Arabia

**Keywords:** coaxial electrospinning, core–shell nanofibers, buccal delivery, fast-dissolving fibers

## Abstract

Geriatric patients are more likely to suffer from multiple chronic diseases that require using several drugs, which are commonly ingested. However, to enhance geriatric patients’ convenience, the electrospun nanofiber system was previously proven to be a successful alternative for the existing oral dosage forms, i.e., tablets and capsules. These nanofibers prepared either as single- or multi-layered fibers could hold at least one active compound in each layer. They might also be fabricated as ultra-disintegrated fibrous films for oral cavity administration, i.e., buccal or sublingual, to improve the bioavailability and intake of the administered drugs. Therefore, in this work, a combination of nifedipine and atorvastatin calcium, which are frequently prescribed for hypertension and hyperlipidemia patients, respectively, was prepared in a coaxial electrospinning system for buccal administration. Scanning electron microscopy image showed the successful preparation of smooth, non-beaded, and non-porous surfaces of the drug-loaded nanofibers with an average fiber diameter of 968 ± 198 nm. In contrast, transmission electron microscopy distinguished the inner and outer layers of those nanofibers. The disintegration of the drug-loaded nanofibers was ≤12 s, allowing the rapid release of nifedipine and atorvastatin calcium to 61% and 47%, respectively, after 10 min, while a complete drug release was achieved after 120 min. In vitro, a drug permeation study using Franz diffusion showed that the permeation of both drugs from the core–shell nanofibers was enhanced significantly (*p* < 0.05) compared to the drugs in a solution form. In conclusion, the development of drug-loaded nanofibers containing nifedipine and atorvastatin calcium can be a potential buccal delivery system.

## 1. Introduction

Geriatric patients aged 60 or older are more likely to suffer from multiple chronic diseases; therefore, they require more attention for their medicines [1]. According to WHO predictions, the number of people aged 60 years and over is expected to reach 1.4 billion by 2030 compared to only a billion in 2020. By 2050, the population of the elderly will reach 2.1 billion. The human body undergoes many physiological changes as they grow older. Several factors affect the nutritional status of older people, such as an insufficient chewing ability due to missing teeth or dentures, slow swallow reflex, difficulty in eating, a decrease in digestive enzymes, a decrease in minerals, vitamins, and drug absorption, and slow digestion rates [2]. Geriatric patients are expected to have poor compliance with the usual drug dosage forms, i.e., tablets and capsules [3,4]. Fortunately, new advanced techniques in pharmaceutical technology have emerged recently to overcome some of the obstacles and limitations of the conventional dosage forms such as the difficulty of swallowing the drugs, and also have presented many oral dosage alternatives for geriatrics, such as buccal drug administration [3,5].

Administering the drug by a buccal route can eliminate several problems, for instance, high first-pass metabolism and drug degradation in the gastrointestinal environment [6]. Various bioadhesive mucosal dosage forms have been developed, such as patches, gels, ointment, and polymeric films for buccal administration, known as mouth dissolving films [5]. A fast-dissolving oral thin film is an ultra-thin film composed of a hydrophilic polymer that rapidly hydrates or adheres to the tongue or buccal cavity [7]. As the film disintegrates and dissolves within seconds, the active agent can be released without drinking or chewing [6,8]. Additionally, since the mucosa is highly enriched with blood supply, drugs would readily be absorbed, enhancing bioavailability [9].

Nevertheless, there are some limitations of orally disintegrating films, including difficulties in achieving dose uniformity and even thickness of the formed films, which affect their mechanical properties and release behavior within the same batch [10]. Consequently, electrospun nanofibers were suggested as an alternative system which has shown considerable potential in controlling the release rate with its unique mechanical properties [11]. Furthermore, electrospun nanofibers have also proved their relatively rapid disintegration that occurs within less than two seconds, compared to other orally disintegrating films, which disintegrate in ≤30 s [10,12].

Electrospinning is a process that produces nanofibers by applying a high voltage on a droplet of a viscous polymer solution to overcome its surface tension, leading to evaporation of the solvent and the formation of a fibrous mat [13]. It is a simple, rapid, inexpensive, and easily scalable technology that can form single- or multi-layered fibers using mono- or multiaxial nozzle systems, respectively, allowing the delivery of multiple drugs in a more controlled-release fashion [13,14,15]. Other electrospinning techniques, such as side-by-side or layer-by-layer electrospinning, could be used alternatively to the coaxial system. They involve spinning two polymer solutions using parallel syringes or one syringe in two different run times, respectively. Amorphous solid dispersion of drugs formed by different electrospinning techniques holds another advantage, particularly for drugs categorized in the Biopharmaceutical Classification System class II (BCS II). This is due to the improvement of the solubility of such drugs, which will, in turn, enhance their release rate and bioavailability [16]. Additionally, electrospun fibers have a high surface-area-to-volume ratio, high porosity, and high drug-loading capacity [17].

Nifedipine and atorvastatin calcium is prescribed frequently in combination as blood pressure-lowering agents and lipid-lowering agents, respectively. Nifedipine is a systemic calcium channel blocker, a yellow crystalline substance practically insoluble in water but soluble in ethanol, that displays significant pharmacological properties and negligible side effects [18,19]. On the other hand, atorvastatin calcium is a member of the lipid-lowering agent and a significant inhibitor of 3-hydroxy-3-methylglutaryl-coenzyme (HMG-CoA) reductase, which catalyzes HMG-CoA conversion to mevalonate at the early stage of cholesterol synthesis [20]. There have been some pieces of evidence that atorvastatin calcium can effectively reduce both cholesterol and triglycerides [18].

The current study aims to fabricate nifedipine and atorvastatin calcium loaded in Polyvinylpyrrolidone (PVP) core–shell nanofibers as a potential buccal dosage form for two BCS class II drugs that are commonly used for geriatrics [20,21]. PVP is a film-forming agent that is a non-ionic, biocompatible, and biodegradable polymer and is known to be safe in food and pharmaceutical applications [22]. Various solid dispersions were prepared using PVP to enhance the dissolution rates of poorly water-soluble drugs in addition to their mucoadhesive properties [17]. This new formula would allow both drugs to dissolve and release into the oral cavity in a few seconds without chewing or drinking. This will, in turn, accelerate the onset of action, avoid first-pass metabolism through oral mucosal delivery, and provide high patient compliance, adherence, and acceptability, especially for geriatric patients who suffer from difficulty swallowing (i.e., dysphagia).

## 2. Materials and Methods

### 2.1. Materials

Polyvinylpyrrolidone (PVP, average molecular weight ~1,300,000), formic acid, ethyl alcohol (≥99.5%) sodium chloride (≥99.5%), sodium phosphate dibasic (≥99.0%), potassium chloride (99.0–100.5%), and potassium phosphate monobasic (≥99.0%), hydrochloric acid HCl (36.5–38.0%), atorvastatin calcium was purchased from Sigma-Aldrich (St. Louis, MO, USA). Acetonitrile was obtained from PanReac AppliChem ITW Reagents (Barcelona, Spain), while nifedipine was purchased from Biosynth Carbosynth (Compton, UK). Deionized water was generated through Milli Q, Millipore (Billerica, MA, USA) and has been used throughout the study. Phosphate buffer saline (PBS) was prepared by mixing 8 g NaCl, 0.2 g KCl, 0.24 g KH₂PO₄, and 1.44 g Na₂HPO₄ in 1 L of distilled water, and the pH was adjusted to 7 using 5M HCL.

### 2.2. Preparation of Nanofibers Using Electrospinning Technique

Drug-loaded spinning solutions were prepared by dissolving PVP in ethanol at 8% (*w/v*) and stirring it at ambient temperature over a magnetic stirrer for at least 60 min (min) until it dissolved completely. 1% (*w/v*) of nifedipine and atorvastatin calcium was added to the PVP polymer solutions (i.e., separately into two separate vials) and stirred for another 60 min to achieve homogeneous solutions. Blank (i.e., drug-free) spinning solution was prepared using only PVP in ethanol at 8% (*w*/*v*) without any drug added.

Spraybase^®^ (Dublin, Ireland) provided a coaxial electrospinning setup to prepare the core–shell fibers using a modified method of (12). Coaxial needles contained inner and outer needles with 0.45 and 0.9 mm diameters, respectively. This needle was attached to two syringes filled with spinning solutions to prepare a core–shell fibrous system. Atorvastatin calcium was injected through the outer needle, whereas nifedipine was injected via the inner needle. The distance between the spinneret and the collector was adjusted to 15 cm, and the collector surface was covered with aluminum foil. The flow rate was maintained at 0.5 mL/h for each layer throughout the whole processing run, whereas the applied voltage was adjusted to 8–9 kV. The entire process was performed under ambient conditions, i.e., room temperature as 20–25 °C and relative humidity as 30–45%. Due to the photosensitivity of nifedipine, electrospinning was performed in a dark room. The blank fibers were fabricated under similar conditions to the drug-loaded fibers but added no drugs.

### 2.3. Scanning Electron Microscopy (SEM)

The morphology and diameters of the fibers were examined and analyzed using SEM (JSM-IT500HR SEM, JEOL Inc., Peabody, MA, USA). Fibers were collected directly onto aluminum foil and observed at an accelerating voltage of 5 kV, and the samples were not coated with any conductive material. The fibers’ diameters were measured using the SEM software (SEM Operation, 3.010, Akishima, Tokyo, Japan), in which the average of at least 15 fibers was measured.

### 2.4. Transmission Electron Microscopy (TEM)

The inner and outer layers of the core–shell fibers were distinguished via TEM (JEM-1400 TEM, JEOL, Tokyo, Japan). Fibers were collected directly onto a copper grid during the electrospinning process, and images were taken at an accelerating voltage of 120 kV.

### 2.5. X-ray Diffraction (XRD)

XRD analysis was conducted using Rigaku miniflex 300/600 (Tokyo, Japan), equipped with a Cu Ka radiation source excitation voltage of 40 kV, and 15 mA current. Each drug, PVP, physical mixture (PM), drug-loaded, and blank fibers were analyzed by placing the sample on the holder. The PM was prepared by weighing the polymer and drugs at equivalent concentrations to the drug-loaded fibers. The data recording was taken between 2θ ranging from 2° to 60° at a scan speed of 5°/min.

### 2.6. Fourier-Transform Infrared Spectroscopy (FTIR)

FTIR analysis was conducted using a Thermo smart ATR IS20 Spectrometer (Thermo Fisher Scientific, Waltham, MA, USA). Small amounts (3–5 mg) of each drug, PVP, PM, drug-loaded, and blank fibers, were placed on the sampling spot. Samples were examined at a wavenumber ranging from 4000 to 600 cm^−1^, with the spectral resolution set at 4 cm^−1^, whereas the number of scans was recorded as 32 for each sample.

### 2.7. Disintegration Test of the Electrospun Fibers

The disintegration of the blank and drug-loaded fibers was assessed using a modified method of the Petri dishes assay described in Tawfik et al. [10]. Previous studies have used the Petri dishes method to evaluate the disintegration of electrospun fibers and other oral dispersible films [12,23,24]. Approximately 4 mg of each fiber mat was placed into 8 mL of pre-warmed PBS (pH 7) under gentle stirring to ensure complete detachment of the fibers. Experimental conditions were conducted in a thermostatic shaking incubator (Excella E24 Incubator Shaker Series, New Brunswick Scientific Co., Enfield, CT, USA) at 37 °C. The measured results represent the mean ± SD of three replicates.

### 2.8. Nifedipine and Atorvastatin Calcium Determination and Quantification Using High-Performance Liquid Chromatography (HPLC)

The HPLC analysis was carried out on a 1260 Infinity II HPLC system (Agilent, Santa Clara, CA, USA) which comprised a binary solvent gradient flexible pump (G7111A), an autosampler (G7129A), and a UV detector (G7114A). The chromatographic separation of nifedipine and atorvastatin calcium was achieved using an Agilent HPLC column Poroshell 120 EC-C18 (4.6 mm × 150 mm, 4 μm) column. Solvents were HPLC grade, and the mobile phase consisted of two solutions, namely A (water + 0.05% formic acid) and B (acetonitrile + 0.05% formic acid). The gradient elution method was as follows: 2 min, 5% B; 8 min, 5–98% B; 1 min, 98% B; 1 min 98–5%, 3 min 5% (B), and a total run time of 15 min and injection volume 10 µL. Nifedipine dissolved with 60% acetonitrile and 40% PBS (pH 7), while atorvastatin calcium dissolved in 80% acetonitrile and 20% PBS (pH 7).

Nifedipine and atorvastatin calcium were detected at retention times of *R_t_* = 11.9 min and 12.2 min, respectively. Calibration curves of both drugs were constructed using a series of concentrations ranging between 0.39 and 200 μg/mL. The result is presented in the Appendix A. The quantification of the drugs was processed using OpenLab CDS software and based on the integrations of the corresponding UV absorption peaks at 335 nm for nifedipine and 245 nm for atorvastatin calcium.

### 2.9. Determination of Drug Loading (DL), Entrapment Efficiency (EE%), and Fiber’s Yield (Y) of the Drug-Loaded Nanofibers

The DL and EE% of the drug-loaded fibers were measured by weighing 8 ± 0.5 mg of fibers and dissolving each sample in 10 mL 80% PBS (pH 7) and 20% acetonitrile solvent mix. Each solution was kept at ambient temperature covered with aluminum foil for six hours to ensure the complete dissolution of the fibers. Following that, the developed HPLC method was used to evaluate the samples, in which the DL and EE% values were determined using the following Equations:(1)$$DL=\frac{Entrapped\;drug\;amount}{Weight\;of\;the\;fibers}$$
(2)$$EE\%=\frac{Actual\;drug\;amount}{Theoretical\;drug\;amount}\times 100\;$$

The theoretical amount of fibers was calculated based on the number of solids (polymer and drugs) in the total volume of the spun solution. Results represent the mean ± SD of at least three replicates.

The Y of the fibers was also measured using the following Equation:
(3)$$Y=\frac{Actual\;amount\;of\;fibers}{Theoretical\;amount\;of\;fibers}\times 100\;$$

### 2.10. In Vitro Drug Release Determination of the Drug-Loaded Nanofibers

Measurement of the drug release was performed by placing approximately 8 mg of the drug-loaded fibers into glass vials containing 20 mL of pre-warmed PBS (pH 7). The vials were kept in a thermostatically shaking incubator at 37 ± 0.1 °C and 100 rpm, according to Alkahtani et al. study [12], but in a dark room. Samples measuring 1 mL were withdrawn after 2, 5, 10, 15, 30, 60, 120, 180, 240, and 300 min and replaced with equivalent volumes of pre-warmed fresh buffer to maintain the sink conditions. The cumulative release % was measured as a function of time and was calculated according to the following Equation:(4)$$Cumulative\;Release\;\%=\frac{Cumulative\;drug\;amount}{Theoretical\;drug\;amount\;}\times 100\;$$

The findings represent the mean ± SD of at least three replicates.

### 2.11. In Vitro Drug Permeability Determination of the Drug-Loaded Nanofibers

To determine the permeability of nifedipine and atorvastatin calcium, a permeability study using a Franz diffusion cell system (PermeGear, Hellertown, PA, USA) was conducted according to Chutoprapat et al. study [25]. Spectra/por 4 dialysis membrane (12–14 kD MWCO) was placed between the donor and receptor compartments of the Franz diffusion cell, and then the compartments were clamped together. PBS (pH 7) was used to fill the donor and receptor compartments. The system was maintained at 37 ± 0.1 °C, while the diffusion medium was stirred at 600 RPM. The open ends of the apparatus with parafilm and aluminum foil to prevent the solvent from evaporation and the drug from being degraded by light.

The system was running for 15 min to equilibrate, and then the donor fluid was replaced with a pre-warmed PBS containing 1 mg of each drug, either as a solution or a fiber. The solution was prepared by mixing 1 mg of each drug in 0.5 mL of acetonitrile (≈25% *v*/*v*) and 1.5 mL of PBS (≈75% *v/v*) until it completely dissolved. At predetermined time points (0.5 h, 1 h, 2 h, 4 h, 6 h, and 24 h), a 100 µL sample was withdrawn from the receptor compartment and replaced with fresh pre-warmed buffer to maintain sink conditions. The cumulative release % was measured according to Equation (4). The results represent the mean ± SD of three replicates.

### 2.12. In Vitro Cytotoxicity Assessment of Nifedipine and Atorvastatin Calcium

The in vitro cytotoxicity evaluation of the loaded drugs, nifedipine, atorvastatin calcium, and their combination (in a 1:1 ratio) was conducted using the MTS assay, following a modified method of Alkahtani et al. [26]. The MTS reagent (cell Titer 96 ^®^Aqueous one solution cell proliferation Assay) was supplied by Promega (Southampton, UK). The experiment was performed after incubating the loaded drugs with the human fibroblast HFF-1 cells (ATCC-SCRC-1401) for 24 h. As previously reported, these cells are often used alternatively in the oral mucosal cavity drug delivery research [27]. The living cellular model was used between passages 25 and 35. The culturing of human cells was routinely maintained in Dulbecco’s modified eagle medium (DMEM), supplemented with 10% (*v/v*) fetal bovine serum (FBS), penicillin 100 U/mL, and streptomycin 100 μg/mL, all purchased from Sigma-Aldrich (St. Louis, MO, USA).

The cells were harvested using trypsin and counted with the trypan blue exclusion test, followed by seeding into 96 well plates at a seeding density of 1 × 10^4^ cells/well. The cells were incubated overnight in a humified 5% CO_2_ cell culture incubator at 37 °C. 100 µL of increasing concentration of the tested compounds (from 15.6 to 500 µg/mL) were then incubated with the human cells for 24 h. Cells incubated with 0.1% triton x-100 were used as the negative control, whereas cells incubated with DMEM only were used as the positive control. Subsequently, samples were removed from the wells, and 100 µL of DMEM was added, followed by adding 20 µL of the MTS reagent into each well. Next, the cells were incubated for 3 h at 37 °C. MTS absorbance was measured at 490 nm using Cytation 3 absorbance microplate reader (BIOTEK instruments inc, Winooski, VT, USA). Finally, the cell viability percentage was calculated using the following Equation:(5)$$Cell\;Viability\;\%=\frac{\left(S-T\right)}{\left(H-T\right)\;}\times 100\;$$
where *S* is the absorbance of the cells treated with the applied drugs, *T* is the absorbance of the cells treated with triton x-100, and *H* is the absorbance of the cells treated with DMEM. The results represent the mean ± SD of three replicates.

### 2.13. Statistical Analysis

The disintegration, DL, EE%, Y, drug release, and in vitro drug permeation studies were carried out in at least three independent replicates, and the findings were presented as mean value ± SD. The data were analyzed using OriginPro 2016 software (OriginLab Corporation, Northampton, MA, USA) for the disintegration, regression equation, correlation coefficient (R^2^), DL, EE%, Y, drug release, XRD, and FTIR studies. The in vitro drug permeation study was analyzed using a t-test through GraphPad Prism^®^ statistical software, and *p* < 0.05 was taken as a criterion for a statistically significant difference.

## 3. Results and Discussion

### 3.1. Fibers’ Morphology and Diameter Analysis

The SEM image showed that drug-loaded fibers had smooth, non-beaded, and non-porous surfaces with an average diameter of 968 ± 198 nm, as shown in Figure 1. Additionally, no drug crystals were observed on the surface of these fibers, indicating the successful integration of this drug into the fibers. These morphological features represent successful fiber preparatory criteria according to [15]. The drug-loaded fibers were also observed under the TEM. The image illustrated two distinct layers, representing the inner and outer layers of the core–shell nanofibrous system, as demonstrated in Figure 2.

### 3.2. X-ray Diffraction (XRD)

The physical state of the raw materials, i.e., PVP, nifedipine, atorvastatin calcium, and their PM, as well as the blank and drug-loaded fibers, were examined using XRD, as shown in Figure 3. The pattern of both raw materials, nifedipine, and atorvastatin, shows a sequence of intense Bragg reflections. The pattern of nifedipine showed a sequence of intense Bragg reflections at 2θ: 8°, 15°, 19°, 21°, and 25°, which is consistent with da Costa et al. [28], Lodagekar et al. [29]. At the same time, atorvastatin calcium demonstrated a sequence of narrow and intense Bragg reflections at 2θ: 2°, 5°, 9°, 10°, 11°, 19°, 21°, and 43°, which is in agreement with a previous study of Choudhary et al. [30]. The same reflections of both drugs could be seen in the diffractogram of the PM, suggesting their presence in a crystalline form within the polymer mixture. On the contrary, PVP polymer showed no intense characteristic peaks, which appeared as a broad halo diffraction pattern. The lack of such reflections indicates the amorphous state of this polymer, which is consistent with [12,31].

The blank and drug-loaded nanofibers diffractograms resembled the PVP polymer, which showed no intense characteristic Bragg reflections of the crystalline drugs and only broad haloes. This result suggested that both nanofiber systems are in an amorphous state Zhao et al. [32]. The change in the physical form of the drugs after electrospinning agreed with the previous studies [12,31]. This transformation can be due to the rapid evaporation of the solvent that prevented the organization of molecules into the crystalline lattice and led to the propagation of a disordered arrangement of molecules in the fibrous mat, as explained in Bukhary et al. [23].

### 3.3. Fourier-Transform Infrared Spectroscopy (FTIR)

The FTIR spectroscopy was conducted to evaluate the compatibility between the drugs and the polymer, i.e., drug-polymer intermolecular interactions, to ensure the resultant formulation’s stability. The FTIR spectrum of the pure PVP polymer, atorvastatin calcium, nifedipine, and their PM showed characteristics peaks of each pure raw material, as shown in Figure 4. Broadband of O–H stretching was observed at 3419 cm^−1^, owing to the hygroscopic nature of PVP, while peaks located at 1651, 1421, and 1285 cm^−1^ were corresponding to the stretching vibrations of C = O, C–H (in the aliphatic compound), and C–N (in aromatic amine), respectively. This spectrum is consistent with [12,31].

The FTIR spectrum for atorvastatin calcium demonstrated stretching at the function group region representing hydrogen bond stretching (–OH group 3668 cm^−1^), amine group stretching (–N–H 3362 cm^−1^), asymmetric O–H stretching at 3230 cm^−1,^ and symmetric O–H stretching at 3054 cm^−1^. Additionally, characteristic bands at 1649 cm^−1^ (asymmetric C = O stretching), 1577 cm^−1^ (symmetric C = O stretching), 1214 cm^−1^ (C–N stretching/C–O stretching), and between 1550–1430 cm^−1^ (four bands related to C–C ring stretching) were also shown in the spectrum of this drug, which is in agreement with Lemsi et al. [33]. The IR spectrum of pure nifedipine showed peaks at 3324 cm^−1^ (N-H stretching), 1675 cm^−1^ (ester carbonyl stretching band), 1646 cm^−1^ (stretching vibration of C = C in the aromatic ring), 1118–1120 cm^−1^ (ether absorption bands of C3 and C5, respectively), and 1308 and 1525 cm^−1^ (nitro group). This spectrum was also seen in the previous study of Jagtap et al. [34].

The stretching peaks at 3410–3420 cm^−1^ and 1651 cm^−1^ were observed in the PM, blank, and drug-loaded nanofibers, suggesting the presence of PVP polymer, as shown in Figure 4. However, since the stretching band at 1493 cm^−1^ and 1509 cm^−1^ appeared in the IR spectrums of both drugs, the PM and drug-loaded nanofibers, while it was absent in the PVP and blank nanofibers spectrums, this band can be an indication for the presence of the drugs in the drug-loaded nanofibers (Figure 4).

### 3.4. Disintegration Test of the Drug-Loaded Nanofibers

When dosage forms are designed to be used in the mouth, disintegration time is an essential qualitative factor [12]. The United State Food and Drug Administration (USFDA) recommendations state that non-chewable and non-liquid drug products should disintegrate within ≤30 s (s) [35]. Both blank and drug-loaded fibers were observed to be disintegrated rapidly in PBS (pH 7). As shown in Figure 5, the blank nanofibers disintegrated and dissolved more rapidly (≤2 s) than the drug-loaded nanofibers (≤12 s). In addition, the disintegration of the blank nanofibers initiated from the center of the fibrous mat to reach the edges.

In contrast, the drug-loaded nanofibers started to disintegrate from the edges to the center, causing the shrinkage of the fibrous mat as a gel-like material until it dissolved completely. This longer disintegration time of the drug-loaded nanofibers might be due to the hydrophobicity of nifedipine, which can be seen in Figure 5b, since the drug has a yellow color. Nevertheless, the nanofibers exhibited a less compact matrix that allowed air pockets to form within the fibrous mat, permitting liquid to be absorbed faster by capillary action [36]. Therefore, these drug-loaded core–shell nanofibers can be considered a suitable delivery system for buccal administration of nifedipine and atorvastatin calcium, particularly for elderly patients.

### 3.5. Drug Loading (DL), Entrapment Efficiency (EE%), and Fiber Yield (Y) of the Drug-Loaded Nanofibers

The DL and EE% of the drug-loaded core–shell nanofibers were measured to be 62 ± 6 µg/mg and 111 ± 11% for nifedipine and 54 ± 2 µg/mg and 97 ± 4% for atorvastatin calcium, respectively, which was determined by the developed HPLC method, as shown in Appendix A. This very high EE% (>95%) was also achieved in the core–shell fiber systems of (28, 29). The Y of the drug-loaded nanofibers was measured as 99 ± 4%, with the remaining amount most likely lost while peeling off the fibers from the aluminum foil. Overall, these findings can promote the successful preparation of this nanofibrous system.

### 3.6. In Vitro Drug Release Determination of the Drug-Loaded Nanofibers

The drug-loaded coaxial system showed a burst release in the initial 10 min for nifedipine and atorvastatin calcium, followed by >80% cumulative release after 30 min, and a complete drug release after 120 min, as shown in Figure 6. This result was expected owing to the hydrophilic nature of PVP polymer, which accelerated the fibrous-mat disintegration and dissolving and, thus, the release of the loaded drugs. This finding is also consistent with Aburayan et al. [31], Sriyanti I et al. [37], Moydeen et al. [38]. In addition, due to the high surface-to-volume ratio of the nanofibers, the contact area with the dissolution medium was enhanced, allowing the rapid release of the drugs [12]. Another factor that enhanced the release of both drugs, categorized as BCS class II, is the molecular dispersion of both drugs upon electrospinning, which was previously confirmed in Section 3.2. [16].

### 3.7. In Vitro Drug Permeability Determination of the Drug-Loaded Nanofibers

The permeability of nifedipine and atorvastatin calcium was evaluated after being formulated by electrospinning, using a Franz diffusion cell method. Nifedipine and atorvastatin calcium are classified as BCS class II; hence they have high permeability and low solubility. Due to the poor solubility of the drugs in PBS, a PBS solution containing 25% *v/v* acetonitrile was used to dissolve both drugs (in equal concentrations) to assess their permeability. As shown in Figure 7, the permeation of nifedipine and atorvastatin calcium was significantly enhanced (*p* < 0.05) compared to the control (i.e., drugs in solution form) at different time points, in which it showed a sustained drug diffusion pattern. This difference was probably because of the molecular dispersion of both drugs in the nanofibers that improved the solubility of the drugs in PBS, and consequently, the permeability was enhanced through the artificial membrane [25,39]. Additionally, since PVP has a high mucoadhesion property, drugs could retain on the membrane for a longer duration, enhancing the permeability [40].

### 3.8. In Vitro Cytotoxicity Assessment of Nifedipine and Atorvastatin Calcium

The in vitro cytotoxicity evaluation of any drug or formulation is essential to assess the safety profile of the administered medications as the first step toward the biomedical application and prior to the therapeutically use in order to avoid any harmful effect on the living tissues. In the present work, Atorvastatin calcium, nifedipine, and their combination (in a 1:1 ratio) were exposed to HFF-1 cells, as shown in Figure 8. When increasing its dose, the application of atorvastatin calcium alone demonstrated a significant decrease (*p* < 0.05) in the HFF-1 cell viability. All concentrations ≥31.2 µg/mL showed cell viability below 50%, while the lowest applied concentration (15.6 µg/mL) exhibited high cell viability (≈100%). On the other hand, Nifedipine demonstrated a very safe profile upon the 24 h exposure on HFF-1 at all concentrations. The combination of atorvastatin calcium and nifedipine in a 1:1 ratio showed only one safe concentration point, at 15.6 µg/mL, and all higher concentrations were toxic to this cell line (cell viability <20%). These findings suggested that more caution should be taken when administered these drugs as a buccal dosage form since this route of drug administration can enhance the bioavailability of drugs by bypassing the first-pass metabolism directly to the bloodstream.

## 4. Conclusions

Drug-loaded core–shell nanofibers containing nifedipine and atorvastatin calcium were successfully formulated and characterized. The SEM image demonstrated that the fibers had smooth, non-beaded, and non-porous surfaces with an average diameter of 968 ± 198 nm, while the TEM showed two distinct layers of the inner and outer layer, indicating the successful preparation of a core–shell nanofibrous system. The DL and EE% of nifedipine and atorvastatin calcium were 62 ± 6 µg/mg and 111 ± 11%, and 54 ± 2 µg/mg and 97 ± 4%, respectively. This high EE% is another indicator for the successful preparation of this nanofibrous system. The in vitro drug release study of the drug-loaded nanofibers exhibited that more than 80% of both drugs were released after 30 min, and a complete drug release was achieved after 120 min. This accelerated drug release profile was probably due to the molecular dispersion of nifedipine and atorvastatin calcium upon electrospinning, as confirmed in the XRD results. The in vitro permeation study also showed that both drugs were significantly enhanced (*p* < 0.05) compared to the drugs in solution form.

In conclusion, the development of drug-loaded nanofibers containing nifedipine and atorvastatin calcium could be a promising alternative to conventional oral dosage forms, such as tablets and capsules. This system is convenient as it can be administered orally as buccal films, which would increase medications’ compliance, adherence, and acceptability for patients, particularly among geriatrics who suffer from dysphagia. However, in future research related to this work, more in vitro and in vivo studies of several buccal cell lines such as TR146 cells should take place, in order to evaluate the safety of this drug-loaded nanofibrous system when attached to the buccal mucosa before it reaches the bloodstream.

## Figures and Tables

**Figure 1 pharmaceutics-14-00358-f001:**
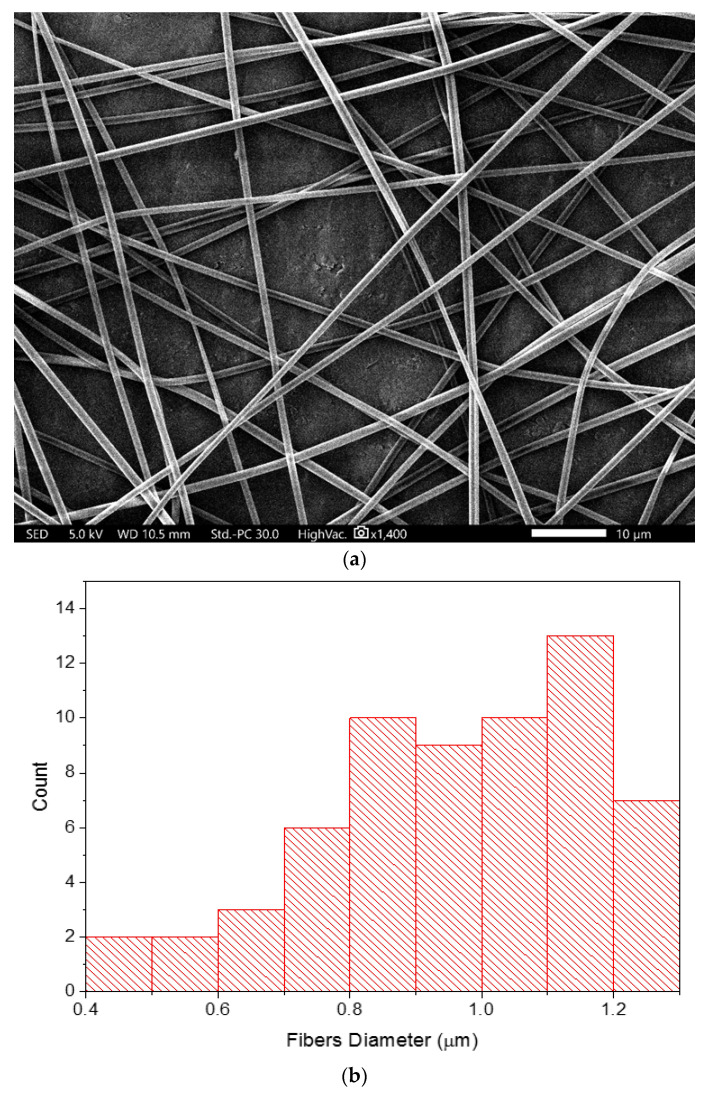
SEM image (**a**) and fibers diameter distribution (**b**) of the drug-loaded core–shell nanofibers, showing smooth, non-beaded, and non-porous surfaces of the fibers, with an average diameter of 968 ± 198 nm.

**Figure 2 pharmaceutics-14-00358-f002:**
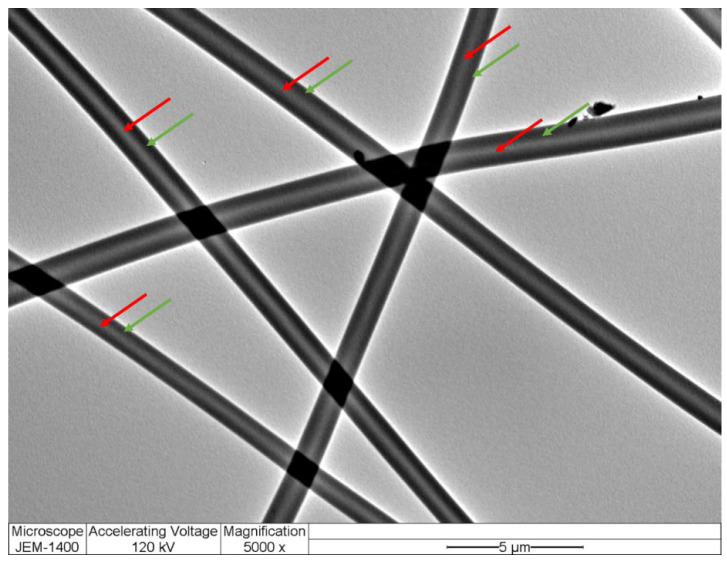
TEM image of drug-loaded core–shell nanofibers showing two distinct layers. The red and green arrows represent the inner and outer layers of the nanofibers, respectively.

**Figure 3 pharmaceutics-14-00358-f003:**
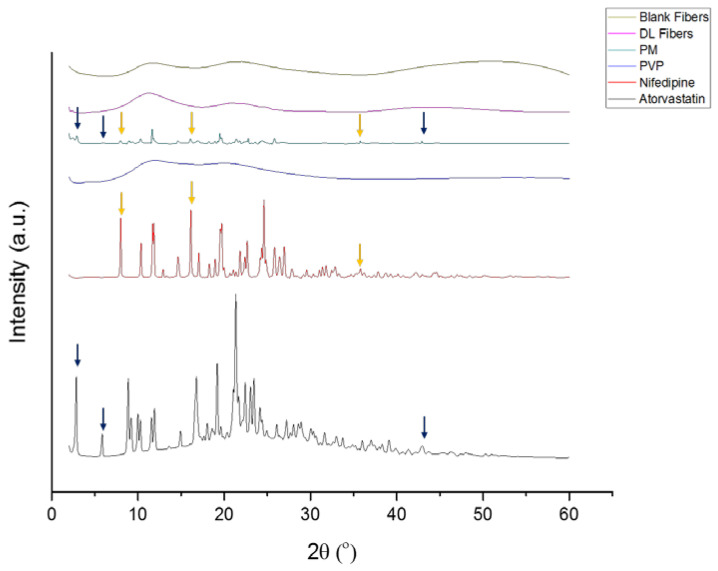
XRD patterns of PVP, nifedipine, atorvastatin calcium, PM, blank and drug-loaded nanofibers showing that both drugs were in the crystalline form (presence of characteristic reflections), while PVP was in the amorphous form (broad halo pattern). The presence of the drugs’ distinct peaks was presented in the PM but was absent in the drug-loaded nanofibers, suggesting the molecular dispersion of the drugs due to the electrospinning process.

**Figure 4 pharmaceutics-14-00358-f004:**
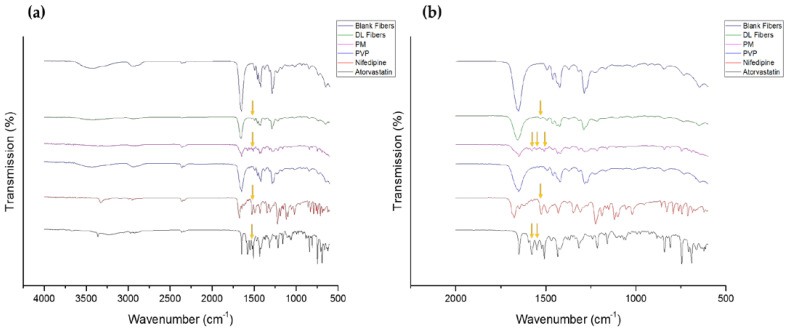
FTIR transmissions at (**a**) a full wavenumber range (4000 to 600 cm^−1^), and (**b**) more focused range (2000 to 600 cm^−1^) of PVP, nifedipine, atorvastatin calcium, PM, blank and drug-loaded nanofibers showed the distinctive drugs’ peaks at 1493 cm^−1^ and 1509 cm^−1^ that appeared in the PM and drug-loaded nanofibers but not in the blank nanofibers.

**Figure 5 pharmaceutics-14-00358-f005:**
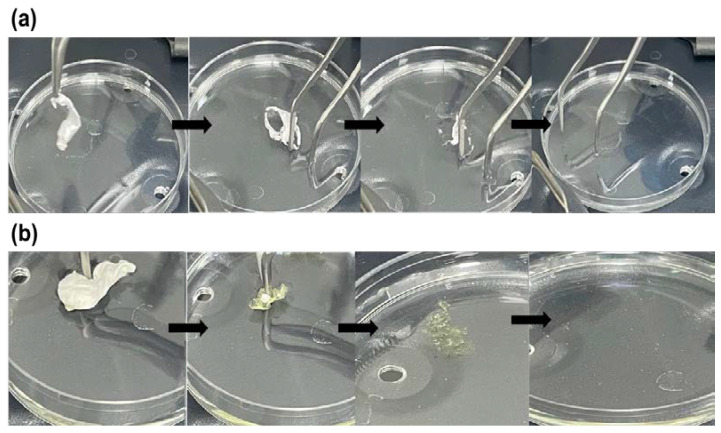
The disintegration of (**a**) blank fibers and (**b**) drug-loaded fibers shows that the blank fiber disintegrated and dissolved more rapidly (≤2 s) than the drug-loaded fibers (≤12 s). The yellow color in the drug-loaded nanofibers indicates the encapsulation of nifedipine.

**Figure 6 pharmaceutics-14-00358-f006:**
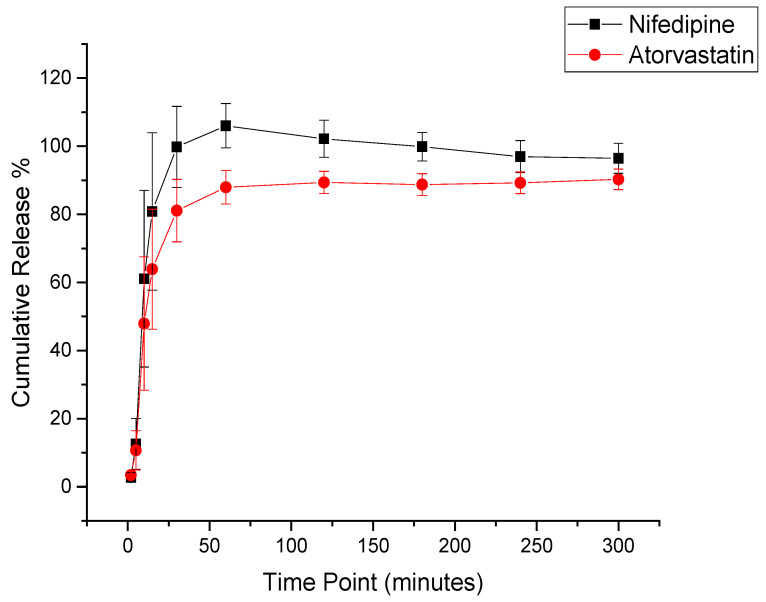
The release profile of the drug-loaded core–shell fibers of nifedipine and atorvastatin calcium showed a burst release at 10 min, followed by >80% release after 30 min and a complete drug release after 120 min—results represented as mean ± SD (*n* = 3).

**Figure 7 pharmaceutics-14-00358-f007:**
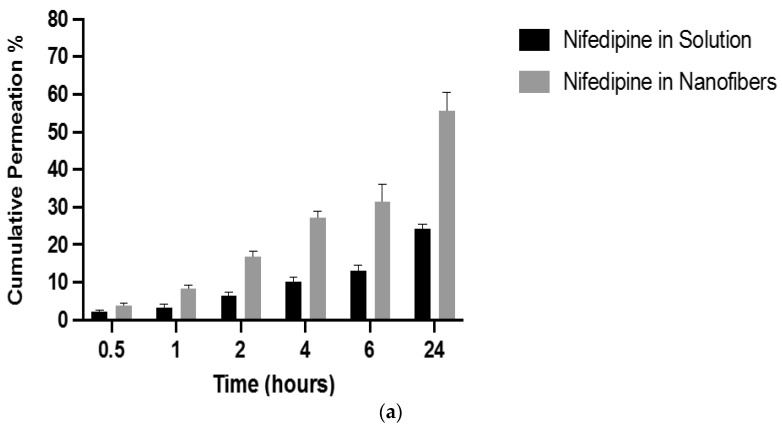
The permeability of (**a**) nifedipine and (**b**) atorvastatin calcium in drug-loaded nanofibers and in drug solution form. The results showed a significantly enhanced permeability (*p* < 0.05) for both drugs from the nanofibers compared to the solution form—results represented as mean ± SD (*n* = 3).

**Figure 8 pharmaceutics-14-00358-f008:**
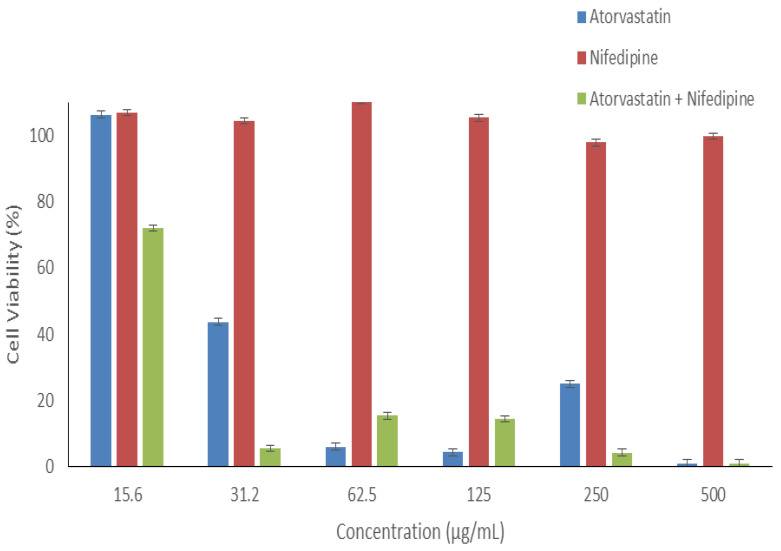
Cell viability of different concentrations of atorvastatin, nifedipine, and their 1:1 combination upon 24 h exposure with HFF-1 cells. The data showed that nifedipine is safer (≤500 µg/mL) than atorvastatin calcium (<30 µg/mL), while it is only safe <15 µg/mL for the combination—results represented as mean ± SD (*n* = 3).

## Data Availability

The authors confirm that the data supporting the findings of this study are available within the article.

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
