# Peer review of "Fast-Dissolving Nifedipine and Atorvastatin Calcium Electrospun Nanofibers as a Potential Buccal Delivery System"

_pharmaceutics, 2022, doi:10.3390/pharmaceutics14020358_

Round 1

Reviewer 1 Report

Comments to the Authors

 Manuscript number Pharmaceutics-1562712-peer-review-v1 “Fast Dissolving Nifedipine and Atorvastatin Calcium 2 Electrospun Nanofibers as a Potential Buccal Delivery System 3 for Geriatrics with Dysphagia” presents the preparation of coaxial electrospun nano-fibers systems for buccal administration of nifedipine in combination with atorvastatin calcium, drugs that are usually prescribed for hypertension and hyperlipidemia to old patients. SEM images demonstrated the good preparation non-porous surfaces of the drug-loaded nanofibers with an average fibers’ diameter of 887 ± 49 nm, while TEM images separated the inner and outer layered of those nanofibers.

In vitro drug permeation study displayed that the permeation of both drugs from the coaxial nanofibers was improved (P < 0.05) as compared with the drugs in a solution form. This study of drug-loaded coaxial nanofibers preparation, containing nifedipine and atorvastatin calcium can be useful as a potential dosage form for elderly patients (with difficulty of swallowing).

The disintegration of the drug-loaded nanofibers was ≤ 12 seconds making possible the rapid release of nifedipine and atorvastatin calcium to 61% and 47%, respectively, after 10 minutes. A complete drug release was achieved after 120 minutes.

The manuscript should have some minor revision before publishing.

Here are some advices for the authors:

-There are some spelling mistakes.

-Fig. 4 can be improved. (Maybe it’s better to increase the 1800-600 zone and represent the relevant peaks in the figure)

-More details about cell viability at the last four values of concentrations (62.5-500). Why these differences? Some comments.

The study is interesting and if the manuscript would have some minor revision before publishing, it will be interesting for the readers of the Journal of Pharmaceutics.

Author Response

Thank you for your valuable comments. Please find the attached response. 

Reviewer 2 Report

Acept i a pesent form. good job

Author Response

Thank you

Reviewer 3 Report

The study reports a fast dissolving buccal film, which is prepared using a coaxial electrospinning process and the core-sheath nanofibers contain two active ingredients. HPLC is utilized for quantitatively determinations. These contents are interesting. I recommend its acceptance for publication in PHARMACEUTICS after some revisions. The following issues please be addressed.

 The title can be “Electrospun Fast Dissolving medicated Nanofibers for Potential Buccal Delivery”. The “Geriatrics with Dysphagia” should be deleted because there are no real investigations on this point. The applications of buccal films can be “Geriatrics with Dysphagia”, babies and also some cases that water is not available.

  “Coaxial Nanofibers” in keywords is not a right phrase. The working process is a coaxial process due to the concentric spinneret and the sheath and core capillaries have a common axial. However, no one can ensure that the core and sheath sections have a common axial after the numerous times drawing. The right word should be “core-sheath” or “core-shell” nanofibers. Please revise all over the texts.

 The fourth paragraph about electrospinning doesn’t give a perfect state-of-art explanation about its development. To make sense, it may be, for example, electrospinning is fast developing from a single-fluid blending process, to coaxial (10.1007/s42114-021-00389-9), side-by-side, tri-axial ( 10.1016/j.jallcom.2020.156471), and other complicated processes . These processes are useful for preparing polymer-based composites, nanostructures such as core-shell , Janus , fiber-bead hybrids, tri-layer core-shell and other hierarchy nanostructures. Correspondingly, amorphous solid dispersion, as a drug-polymer composite, can be created using different kinds of electrospinning processes and exist in various structural formats. Particularly for those BCS II poorly water-soluble drugs.

 How about the sinking conditions of the two drugs.

 Only 7 of the 39 references are within the most recent 3 years, often the standard is 25%. To relate your job with the most recent developments can benefit a high impact of your article after publication. Meanwhile, the formats of the references should be unified.

Author Response

(The authors gave the same response as above.)

Reviewer 4 Report

The submitted draft is a well thought out and structured work; it follows a clear and well-established methodology. However, it generates some doubts that should be taken into account to be publishable.

This work supported by 14 authors seems to me to be excessive the number of participants in which the contribution of each one should be detailed.

As for the written work, there are some typographical errors and some distinctions that should be clarified:

Page 3/15,

-line 103. The authors must be cited after "generally regarded as safe". (X)

-line 122. To prepare the coaxial fibers, you use a coaxial needle with two different dilutions corresponding to each drug in their respective syringes. However, you only use a single flow rate. This assumes that for each of the solutions there is a separate pump for each solution at the same rate or two syringes with the same pump at a single flow rate, therefore, it is not clear in the description of the equipment and the preparation of the coaxial fibers.

-line 147. To make a rigorous measurement assumes at least 100 fibers in different planes, with only 15 samples it is not considered meaningful. A histogram would be preferable to an average measurement of 15 measurements.

Page 6/15, line 262. You use the same tool to formulate equation 5.

Page 7/15. In my opinion, I do not think that Figure 2 corresponding to coaxial fibers by TEM demonstrates that they are coaxial since the effect may be due to the contrast of a cylindrical shape and not to the minimal difference of two drugs that should not present a very different phenomenology. If any of the drugs were fluorescent, it could be seen by microscopy. To demonstrate that the fiber is coaxial, the diameter of a fiber (perpendicular plane of cut fiber) must be visualized in order to confirm this fact.

Page 9/15

Regarding Figure4, it would be preferable to show an enlarged area of the IR transmission spectrum where it can be recognized that the fibers with the drugs at least have bands characteristic of each one of them, but nothing is visible in what they show.

Page 10/15

Section 3.5 should go to a supplementary part. In my opinion, this result is complementary and not fundamental.

Page 12/15

The diffusion figures are not well represented the numerical axes when representing time, such a time scale is not correct, they should be shown as a bar chart instead or with appropriate scalable values (log, etc....). -These figures were poorly represented, and would normally respond to liberation kinetics.

Author Response

(The authors gave the same response as above.)

Round 2

Reviewer 4 Report

I agree with the changes introduced by the authors.